# Role of Telomere Length in Survival of Patients with Idiopathic Pulmonary Fibrosis and Other Interstitial Lung Diseases

**DOI:** 10.3390/biomedicines11123257

**Published:** 2023-12-08

**Authors:** Sofía Tesolato, Juan Vicente-Valor, Jose-Ramón Jarabo, Joaquín Calatayud, Melchor Sáiz-Pardo, Asunción Nieto, Dolores Álvaro-Álvarez, María-Jesús Linares, Carlos-Alfredo Fraile, Florentino Hernándo, Pilar Iniesta, Ana-María Gómez-Martínez

**Affiliations:** 1Department of Biochemistry and Molecular Biology, Faculty of Pharmacy, Complutense University, Ramón y Cajal Sq. (University City), 28040 Madrid, Spain; sofiteso@ucm.es (S.T.); juanvicen@ucm.es (J.V.-V.); 2San Carlos Health Research Institute (IdISSC), 28040 Madrid, Spain; jjarabo@ucm.es (J.-R.J.); jcalatay@med.ucm.es (J.C.); carlosalfredo.fraile@salud.madrid.org (C.-A.F.); florhern@ucm.es (F.H.); anamagom@ucm.es (A.-M.G.-M.); 3Department of Surgery, Faculty of Medicine, Complutense University, Ramón y Cajal Sq. (University City), 28040 Madrid, Spain; 4Thoracic Surgery Service of the San Carlos Hospital, 28040 Madrid, Spain; 5Pathological Anatomy Service of the San Carlos Hospital, 28040 Madrid, Spain; msaizpar@ucm.es; 6Pulmonology Service of the San Carlos Hospital, 28040 Madrid, Spain; asuncion.nieto@med.ucm.es; 7Pulmonology Service of the Mostoles University Hospital, 28935 Madrid, Spain; doloalvaro@yahoo.es; 8Pulmonology Service of Alcorcon Foundation University Hospital, 28922 Madrid, Spain; mjlinaresa@gmail.com

**Keywords:** interstitial lung diseases, idiopathic pulmonary fibrosis, telomere length, prognosis

## Abstract

Interstitial lung diseases (ILDs) constitute a group of more than 200 disorders, with idiopathic pulmonary fibrosis (IPF) being one of the most frequent. Telomere length (TL) shortening causes loss of function of the lung parenchyma. However, little is known about its role as a prognostic factor in ILD patients. With the aim of investigating the role of TL and telomerase activity in the prognosis of patients affected by ILDs, we analysed lung tissue samples from 61 patients. We measured relative TL and telomerase activity by conventional procedures. Both clinical and molecular parameters were associated with overall survival by the Kaplan–Meier method. Patients with IPF had poorer prognosis than patients with other ILDs (*p* = 0.034). When patients were classified according to TL, those with shortened telomeres reported lower overall survival (*p* = 0.085); differences reached statistical significance after excluding ILD patients who developed cancer (*p* = 0.021). In a Cox regression analysis, TL behaved as a risk-modifying variable for death associated with rheumatic disease (RD) co-occurrence (*p* = 0.029). Also, in patients without cancer, ferritin was significantly increased in cases with RD and IPF co-occurrence (*p* = 0.032). In relation to telomerase activity, no significant differences were detected. In conclusion, TL in lung tissue emerges as a prognostic factor in ILD patients. Specifically, in cases with RD and IPF co-occurrence, TL can be considered as a risk-modifying variable for death.

## 1. Introduction

The term interstitial lung disease (ILD) includes an heterogeneous group of more than 200 diseases, characterized by histological and radiological findings of inflammation and/or fibrosis affecting the interstitium of the lungs [1,2]. Although several classifications have been provided, ILDs can be subdivided into autoimmune-related ILDs (e.g., connective tissue disease associated-ILDs (CTD-ILDs)), exposure-related ILDs (e.g., hypersensitivity pneumonitis, drug- or toxin-induced ILDs), ILDs involving cysts or airspace filling, sarcoidosis, orphan diseases and idiopathic ILDs (of which the most common is idiopathic pulmonary fibrosis (IPF)) [3,4]. IPF is a progressively fibrosing ILD with the radiological and/or histological pattern of usual interstitial pneumonia (UIP), in the exclusion of environmental factors ascribed to ILD (such as domestic and occupational environmental exposures, connective tissue diseases or drug toxicity) [5,6]. However, these conditions may be sometimes coetaneous but not judged to be sufficient to exclude the diagnosis of IPF [7]. Estimates of prevalence and incidence of IPF are inaccurate due to the complex nature of diagnosing the disease, but between 7000 and 101,500 people develop the disease annually [8]. Due to this health burden, it is of crucial importance to detect biomarkers that are clinically useful in establishing a differential diagnosis and prognosis between IPF and other forms of progressive pulmonary fibrosis [4,9].

Regarding prognosis, one of the main characteristics affecting the progression of ILDs is the radiological pattern, among which UIP, a definite feature of IPF, confers a worse clinical outcome to affected patients [10,11]. Fibrosis mediators lead to a decline in lung function and are involved in the pathological progression of IPF. Processes such as protein homeostasis imbalance and mitochondrial damage in lung tissue, among others, have exhibited a critical role in the pathophysiology of IPF. Considering molecular factors that may have relevance in the clinical management of IPF patients, previous works have reported that telomere length (TL) and telomerase activity could be impaired in this group of ILDs [12]. Telomeres, DNA sequences located at the end of the eukaryotic chromosomes, protect DNA from progressive shortening during normal cell replication [13]. TL has been implicated in a variety of lung diseases, as their shortening limits the capacity of tissue renewal in the lungs [14,15]. Although the role of telomere shortening as a driving factor for the development of lung diseases such as ILDs has been more widely investigated, little is known about the impact of tissue telomere status on the prognosis of IPF versus other ILDs.

In this work, our main objective was to investigate the role of TL and telomerase activity as possible biomarkers with clinical utility in establishing the different prognosis of patients affected by IPF and those who develop other ILDs. To achieve this aim, we established a translational research work which is schematized as follows (Figure 1):

## 2. Materials and Methods

### 2.1. Patient and Tissue Samples

The study population included 61 patients referred to the Thoracic Surgery Service of the San Carlos Hospital in Madrid (Spain) and submitted to surgical lung biopsy for ILD diagnosis. All participants were recruited subsequently, regardless of gender, age or disease stage.

A total of 28 IPF (7 women and 21 men) and 33 (8 women and 25 men) patients with other ILDs were included prospectively between 2017 and 2023. The median age ± interquartile range (IQR) was 70 ± 9 years for IPF patients and 63 ± 17 years for patients with other ILDs.

Table 1 shows demographic variables (gender, age), co-morbidities and exposures from all the participants considered in this work according to final diagnosis. Both groups were comparable except for age and tobacco exposure, which were significantly higher in the IPF population. 

Written approval to develop this study was obtained from the Clinical Research Ethics Committee of the San Carlos Hospital (C.I. 19/104-E_BS, 26/3/2019). In addition, written informed consent was obtained from patients prior to enrolment. Exclusion criteria were the absence of a lung diagnostic biopsy indication or the lack of signed informed consent.

After surgical resection, tissue samples were frozen until processed at −80 °C, using Tissue-Tek^®^ OCT (Sakura Kinetek USA Inc,. Torrance, CA, USA) as a freezing medium. All the samples were handled by the Biobank of the Health Research Institute of San Carlos Hospital (IdISSC) in Madrid (B.0000725), from the national network of Biobanks, project PT2020/00074, subsidized by the Carlos III Institute of Health (ISCIII) and co-funded by the European Union through the European Regional Development Fund (ERDF). Cryostat-sectioned, haematoxylin and eosin (H&E)-stained samples from each tissue block were examined microscopically by two independent pathologists to confirm pathological diagnosis.

### 2.2. Analysis of Telomere Length and Telomerase Activity

Both relative telomere length (TL) and telomerase activity were analysed in fresh Tissue-Tek^®^ OCT-frozen lung tissue samples. For DNA extraction and protein extraction, 20 cuts of tissue of 10 µm and 10 cuts of 10 µm were used, respectively. In both cases, OCT was removed prior to the extraction by washing the cuts with 500 μL of PBS and centrifuging at 4000 rpm during 10 min at room temperature.

Genomic DNA extraction for TL determination was performed according to the Blin and Stafford procedure [16]. Relative TL in ILD samples was determined by a real-time quantitative PCR (qPCR) method based on the Cawthon procedure [17]. This technique consists of the calculation of the T/S ratio, a relative parameter of the telomere length of the sample that is given by the comparison of the copy number of its telomere sequence (T) between the number of copies of a single copy gene (S), with respect to a reference DNA. As a single copy gene, the RPLP0 (ribosomal protein large P0) gene was used. Then, two real-time qPCR amplifications per sample were performed, one for the quantification of telomeres and the other for the quantification of RPLP0. Each reaction contained 20 ng of DNA, 10 μL of the 2× PowerUp™ SYBR™ Green Master Mix (Thermo Fisher Scientific, Madrid, Spain), which contained ROX™ as a passive reference dye, as well as the corresponding primers for each reaction (Thermo Fisher Scientific), in 20 μL of final volume. The following primer sequences were used [17,18]: TELOMERE 5′-CGGTTTGTTTGGGTTTGGGTTTGGGTTTGGGTTTGGGTT-3′ (forward) and 5′-GGCTTGCCTTACCCTTACCCTTACCCTTACCCTTACCCT-3′ (reverse); RPLP0 5′-CAGCAAGTGGGAAGGTGTAATCC-3′ (forward) and 5′-CCCATTCTATCATCAACGGGTACAA-3′ (reverse).

For the amplification of the telomere sequence, concentrations of 300 nM of the forward primer and 900 nM of the reverse primer were used, and for the amplification of the RPLP0 gene, 300 nM of the forward primer and 500 nM of the reverse primer were used. 

All reactions were performed by triplicate. For each pair of primers, a standard curve was included by using serial dilutions of a pooled human DNA of known concentration obtained from non-tumour lung tissues, in order to calculate the efficiency (E) of each experiment (E=10−1slope, where “slope” refers to the slope of the regression line that relates the Ct value versus the logarithm of the DNA concentration). The same pooled DNA was used as the reference DNA for the T/S ratio calculation. T/S ratio was calculated using the model established by Pfaffl [19]:$$T/Sratio=\frac{{E_{telomeres}}^{\left({Ct}_{reference\;telomeres}-{Ct}_{sample\;telomeres}\right)}}{{E_{RPLP0}}^{{(Ct}_{reference\;RPLP0}-{Ct}_{sample\;RPLP0})}}$$

This T/S ratio constitutes a relative indicator of telomere length. Higher values of the T/S ratio are related to longer telomere sequences.

For telomerase activity determination, total protein extracts were obtained and processed according to the protocol of the TeloTAGGG^™^ Telomerase PCR ELISA kit (Roche Diagnostics, Rotkreuz, Switzerland). This procedure allows a semi-quantitative assay to be established that consists of a telomeric repeat amplification protocol (TRAP)-based telomerase polymerase chain reaction (PCR), followed by an enzyme-linked immunosorbent assay (ELISA) to detect the PCR products. For each sample, assays were performed following the manufacturer’s instructions both in the direct and the 10-fold diluted protein extract. Samples with OD_450nm_ ≥ 0.2 were considered as telomerase positive.

### 2.3. Statistical Analysis

Variables were described using medians and IQR or proportions according to the properties (quantitative or categorical, individually) of the data. Comparisons and statistical tests were performed based on the quantitative (Student’s *t*-test, Mann–Whitney U) or categorical (Chi-squared) nature, and the normality or non-normality of the data, as appropriate. 

The T/S ratio was categorized according to the Cutoff Finder tool [20], which set 0.95 as the cut-off point for classifying patients depending on the TL of their tissue samples. Specifically, patients with lung tissue samples showing a T/S ratio < 0.95 were classified in the group of shorter TL, whereas patients whose tissue samples showed a T/S ratio ≥ 0.95 were classified in the group of longer TL.

Survival curves according to clinico-pathological variables and categorized T/S ratio were obtained based on the Kaplan–Meier method and compared with a log-rank test. Patients who died in the postoperative period (1 month) were excluded from the survival studies. The mean follow-up period of the series was 23 months (range 2–70). 

A Cox regression model was performed to assess the impact on mortality of clinical variables and T/S ratio. Variables to be included in the Cox regression were selected using a step-by approach: first, a univariate Cox test was performed for each variable, and those with statistical *p* < 0.3 were refined by means of backward and forward stepwise selection and the best equation of all possible subsets according to the Harrell’s C, Akaike information criterion (AIC) and Bayesian information criterion (BIC). These strategies allow the selection, from a set of predictor variables, of those that would best fit a regression model [21]. We also tested for the validity of the model (proportionality assumption, log-linear relationship between instant rate of incidence and explicative variables, and presence of influential individuals). The statistical significance value was set at *p* < 0.05. Statistical analysis was performed with STATA IC16.1 (Stata-Corp LLC, TX, USA) and IBM^®^ SPSS^®^ Statistics software package version 27 (IBM Inc., Armonk, NY, USA).

## 3. Results

### 3.1. Survival Analysis Based on Clinico-Pathological Variables and T/S Ratio

When analysing the effect of IPF on the survival of ILD patients (Figure 2), IPF patients experienced more death events (10 (35.7%) vs. 4 (13.3%)) and showed a significantly lower survival time after surgery than patients with other ILDs (estimated Kaplan–Meier probability of death: 100.0% vs. 36.1%, log rank-test *p* = 0.034). The mean T/S ratio was lower in the IPF group, although the difference was not statistically significant (0.762 vs. 0.856, *p* = 0.330 in Student’s *t*-test). Regarding telomerase activity, the proportion of telomerase positivity was higher in the group with higher TL, e.g., the non-IPF group (23 (82.1%) vs. 16 (66.7%), *p* = 0.199 in Chi-squared test). 

Then, we considered the effect of TL on survival of the total population of patients affected by ILD. Specifically, we analysed survival curves according to T/S ratio (Figure 3) and found a higher mortality rate (12 (33.3%) vs. 2 (9.52%)) and poorer survival, with borderline significant differences (estimated Kaplan–Meier probability of death: 75.8% vs. 15.3%; log rank-test *p* = 0.085), in patients with shorter TL (Figure 3). Telomerase activity evaluation reported a greater number of positive cases in those patients with a T/S ratio above the stablished threshold (17 (81.0%) vs. 21 (70.0%), *p* = 0.377 in Chi-squared test).

When considering the effect of TL only on the survival of the group of IPF patients, those patients with shorter telomeres (T/S ratio < 0.95) showed lower survival than the ones with longer telomeres (number of deaths: 8 (42.1%) vs. 2 (25.0%); estimated Kaplan–Meier probability of death: 100.0% vs. 41.7%; *p* = 0.464).

Next, considering the effect of cancer on TL, we established survival analyses according to the categorized T/S ratio in ILD patients with and without lung cancer. Interestingly, in the subpopulation of ILD patients without cancer (n = 36), those with shortened TL had a significantly poor clinical evolution, comparing to the group showing longer TL (number of deaths: 5 [23.8%] vs. 0 [0.0%]; estimated Kaplan–Meier probability of death: 42.9% vs. 0.0%; log rank-test *p* = 0.021) (Figure 4). Telomerase reactivation was also lower in the patients with shortened TL (12 (80.0%) vs. 11 (68.8%), *p* = 0.474 in Chi-squared test). 

Only considering non-cancer IPF patients, survival differences in relation to T/S ratio showed a trend towards statistical significance (Figure 5). Thus, after excluding patients with cancer, subjects affected by IPF and showing shortened telomeres experienced a worse prognosis than those with longer TL (number of deaths: 5 (35.7%) vs. 0 (0.0%); estimated Kaplan–Meier probability of death: 58.4% vs. 0.0%), although differences did not reach statistical significance (*p* = 0.118, log-rank test). In the group of cases with a more adverse prognosis, a slightly lower telomerase activation rate was observed, although without significant differences (7 (63.6%) vs. 3 (60.0%), *p* = 0.889 in Chi-squared test).

Non-IPF patients without cancer did not report any mortality events.

Another clinical variable of interest in the group of pathologies considered in this work is the development of rheumatic diseases (RD). Therefore, we then performed survival analysis according to categorized T/S ratio and stratifying by the co-occurrence of ILD and RD (including sarcoidosis, arthrosis and rheumatoid arthritis, among others) in patients without cancer (Figure 6).

When patients without cancer were stratified based on ILD and RD co-occurrence, the effect of shorter telomeres on survival could be noticed both in patients with and without a RD, although it was only significant in the group of RD patients (number of deaths: 3 (50.0%) vs. 0 (0.0%); estimated Kaplan–Meier probability of death: 73.3% vs. 0.0%, log rank-test *p* = 0.031). Telomerase activity was again decreased in patients with shorter TL (3 (60.0%) vs. 5 (71.4%), *p* = 0.679 in Chi-squared test).

In summary, our results indicate that TL constitutes a relevant factor in establishing the clinical prognosis of patients affected by ILD, with patients having shorter telomeres showing a worse prognosis.

### 3.2. Cox Regression Analysis

In order to assess the impact on mortality of clinical variables and T/S ratio, we performed a Cox regression analysis. The first step of the variable selection process (variables with *p* < 0.3 in univariate Cox regression) yielded as potential prognostic factors age, IPF, RD and cancer, as well as the interactions of T/S ratio with age, IPF and RD. We applied a second selection procedure, by means of stepwise selection and the best equation of all subsets, which proposed age, the presence of IPF, RD, T/S ratio and the interaction between T/S ratio and RD as significant variables. The whole process of variable selection is described in the Appendix A. Except age, the clinical variables selected were implied in prognosis (Table 2). The T/S ratio would behave as an effect modifying variable of RD.

A diagnosis of the selected model (detailed in Appendix A) did not reveal significant disruptions in its assumptions. 

### 3.3. Inflammatory Parameters in Patients with IPF and RD

To explain the modifying effect that the T/S ratio has on the survival impact of RD at a molecular level, we investigated the possible link between shorter telomeres and a higher oxidation status due to inflammation. Thus, we analysed the correlation between TL and several blood inflammatory parameters (fibrinogen, C-reactive protein (CRP), erythrocyte sedimentation rate (ESR) and ferritin) measured at the time that the lung tissue was obtained.

We found that in patients without cancer, ferritin was significantly increased in cases with RD and IPF co-occurrence, when compared to patients with just IPF (*p* = 0.032, Mann–Whitney U test) (Table 3). 

## 4. Discussion

In this work, we evaluated the role of TL and telomerase activity in lung tissue samples of patients affected by ILDs according to the presence of a IPF vs. non-IPF ILDs. As far as we are concerned, this is the first work to evaluate the role of lung tissue TL in the prognosis of patients with ILDs. One previous work used tissue TL as a biomarker for disease development [22], although the majority of studies have focused on peripheral blood leukocyte TL as a biomarker for the presence of ILD [23]. Thus, several other works have shown a worse prognosis in patients with ILDs, such as IPF and CTD-ILD patients, who had shortened telomeres in peripheral white blood cells [24,25,26,27].

We detected a worse clinical evolution in IPF patients, especially in the ones with shortened TL in lung tissues. Specifically, we found an estimated median survival of 66 months in those patients with IPF, similar to that reported elsewhere [24,25], and survival was even lower in those IPF patients with shorter telomeres in lung tissue (28 months). The TL of peripheral blood cells has been proposed as a reflection of the overall TL of the individual [28], including in lung tissue. Nevertheless, considering the absence of inflammatory cells in the lung tissue of IPF patients [29,30,31], it is challenging to elucidate how peripheral leukocyte TL reflects telomere status in lung tissue. Despite this, telomere shortening has been observed to be systemic in patients with IPF [32] and an increased systemic oxidative stress has also been reported [33] in these patients.

Molecular mechanisms have been proposed to explain the involvement of telomere shortening in fibrosis or poor prognosis observed in ILDs with UIP patterns, such as IPF and CTD-ILDs. Telomere shortening in epithelial type II cells and alveolar stem cells, caused by conditions that increase cell turnover, oxidative stress or inflammation (e.g., smoking, mutations and other noxious factors) would lead to a senescence-associated phenotype and influence the native microenvironment and local cell signalling, including increased expression of transforming growth factor-beta (TGF-β) [34]. This mechanism would trigger the deposition of excessive extracellular matrix filaments in the interstitial space and the gradual loss of healthy alveolar epithelial cells in the lung parenchyma. Moreover, when shortened telomeres reach a critical length, cell apoptosis occurs with a further loss of healthy alveolar epithelial type 2 cells and lung dysfunction. In this regard, one study in explanted lungs showed that telomere shortening was associated with increased total collagen and chromosomal damage, and this damage increased elastin and structural disease severity [35]. This mechanism has also been proposed for other diseases, such as asthma and chronic obstructive pulmonary disease. In fact, telomere dysfunction in lung pathophysiology is significant because some diseases, such as IPF, can be familial by inheriting mutations on telomerase reverse transcriptase (TERT) or telomerase RNA component (*TERC*), the regulator of telomere elongation helicase 1 (RTEL1), poly(A)-specific ribonuclease (PARN) implied in maturation, dyskerin (DKC1) implied in trafficking, and TERF1-interacting nuclear factor 2 (TINF2) implied in shelterin function [36,37,38]. 

In our study, we were not able to demonstrate an independent role for TL in the worse prognosis of ILD patients, for several possible reasons. One of the reasons might be differences in the populations, as we included patients with cancer, and the population size was modest. Additionally, we included ILD patients being subsidiary to surgery which might have caused a selection bias. Furthermore, other molecular mechanisms that were not investigated could be involved. Several works highlight the role of the mucin-like protein MUC5B, metalloproteinases MMP7 and MMP9, and interleukins such as CXCL10 and IL-23 in the risk of developing the disease or a more severe disease [39]. Perhaps, when integrating all this information, we could better interpret a patient’s prognosis in a personalized approach. Therefore, we suggest that a panel of molecular markers to assess patient prognosis would enable a more comprehensive approach [40,41]. 

Although telomere shortening did not show an independent role in the risk of death of our population, it did modify the impact that RD had on survival. In fact, in a multivariate analysis, TL displayed a modifying role on the risk of death associated with the presence of RD; additionally, a link between higher inflammatory parameters and shorter TL in patients with IPF and concomitant RD was detected.

Some authors have outlined that CTD-ILDs have a worse prognosis than other ILDs, and similar to IPF, when a UIP pattern is present [42,43]. This emphasizes the importance of combining the radiological and pathological features [44] along with molecular markers when assessing ILD prognosis. Surprisingly, cancer did not impact significantly on mortality in the Cox regression, perhaps due to early stages of the patients at diagnosis, which anticipated a good prognosis. A higher impact of TL in patients with IPF and RD co-occurrence may be explained because the RD exhibit additional mechanisms underlying TL shortening. Our hypothesis is that the greater inflammatory state associated with RD could be the reason that explains the role of TL as a variable that modifies the risk of death in patients with coexistence of IPF and RD. Higher inflammation would lead to increased production of oxygen reactive species, which in turn would damage DNA and shorten telomeres. We conducted a very indirect analysis and showed that patients with concomitant IPF and RD and higher inflammatory parameters tended to have shorter TL. Some studies point at the same direction regarding the involvement of inflammation and short telomeres [45,46,47]. TL has also been reported to have a modifying variable effect in other contexts. For example, in a study assessing the role of TL in the risk of death in IPF patients treated with immunosuppression, the authors showed a significant interaction between immunosuppression medication (e.g., azathioprine, prednisone and N-acetylcysteine) and TL in peripheral leukocytes [48]. Therefore, the authors identified leukocyte TL as a biomarker for patients with IPF at risk for poor outcomes when exposed to immunosuppression. Similarly, we identified tissue TL as a biomarker for patients with ILD at risk of poor outcomes when exposed to concomitant CTDs.

Despite the role of TL in the prognosis of ILD established in observational studies, difficulties in implementation of TL as a prognostic marker in clinical practice may be due to the lack of consensus on the method to be used in the measurement of TL [23,38] and the lack of cutoff value to define shortened TL. Further investigation is warranted to standardize TL as a clinical biomarker.

Our work has several limitations and strengths. First, our sample size is modest and different follow-ups in subgroups may have hampered drawing a statistical conclusion. However, we tested assumptions for Cox regression and found no significant vulnerabilities. Second, the classification of patterns and final diagnosis of ILD is difficult, but the final classification was performed within a multidisciplinary approach. Furthermore, we analysed tissue samples instead of leukocyte TL, which may more accurately reflect the telomere status of lung tissue.

## 5. Conclusions

In conclusion, our results indicated a worse prognosis in IPF patients, especially in the ones with shortened TL. In a multivariate analysis, TL displayed a modifying effect on the risk of death associated with IPF and RD co-occurrence. Specifically, shorter telomeres in lung tissues from that group of patients conferred the worst clinical evolution. Moreover, this group of subjects showed higher levels of inflammatory parameters, which indicates that the state of inflammation associated with RD could explain the death-modifying effect associated to telomere shortening in patients affected by IPF and RD.

## Figures and Tables

**Figure 1 biomedicines-11-03257-f001:**
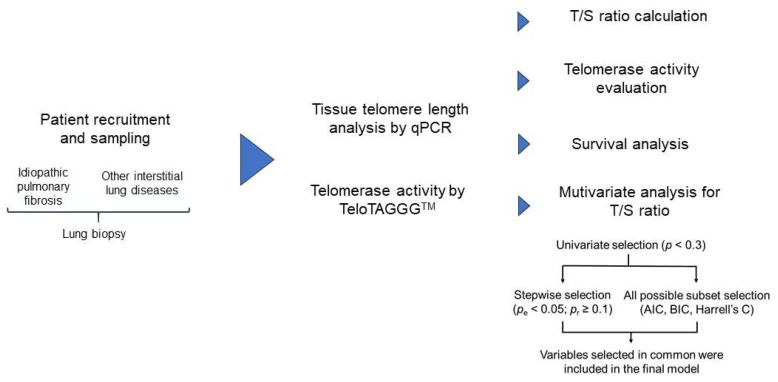
AIC: Akaike information criterion; BIC: Bayesian information criterion; *p*_e_: *p* value for variable inclusion in the stepwise selection process; *p*_r_: *p* value for retrieval of the variable during the stepwise selection process.

**Figure 2 biomedicines-11-03257-f002:**
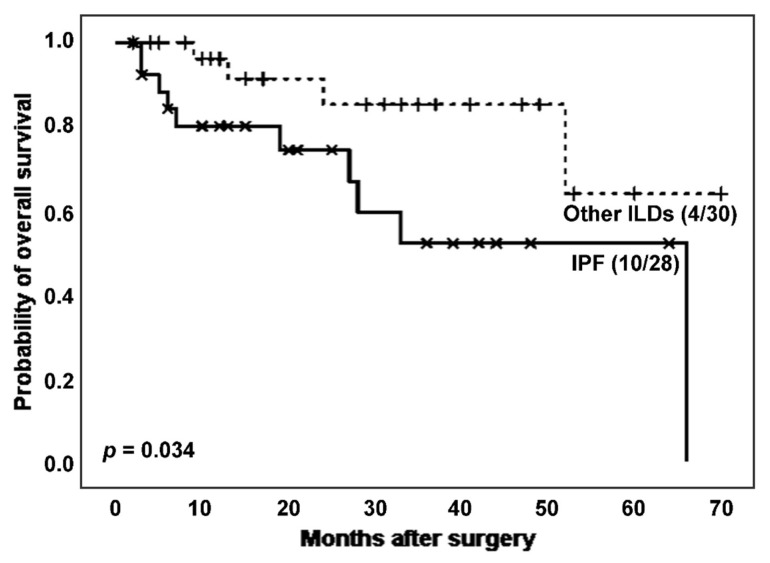
Survival curves according to the presence of IPF vs. other ILDs.

**Figure 3 biomedicines-11-03257-f003:**
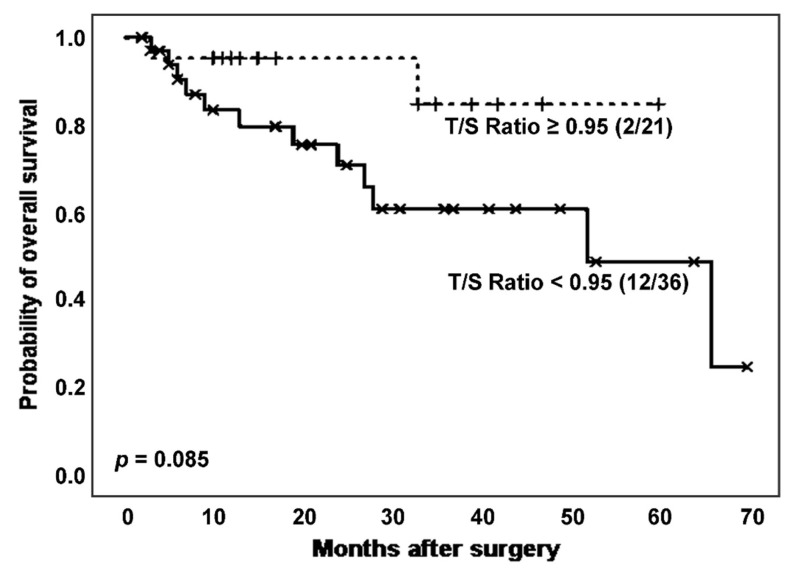
Survival curves of patients affected by ILD according to T/S ratio categorization.

**Figure 4 biomedicines-11-03257-f004:**
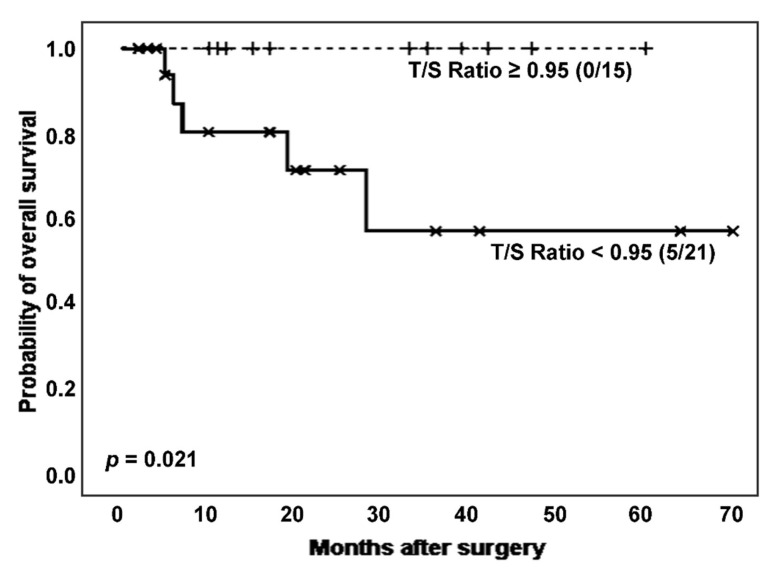
Kaplan–Meier survival curves for ILD patients without cancer, according to T/S ratio categorization.

**Figure 5 biomedicines-11-03257-f005:**
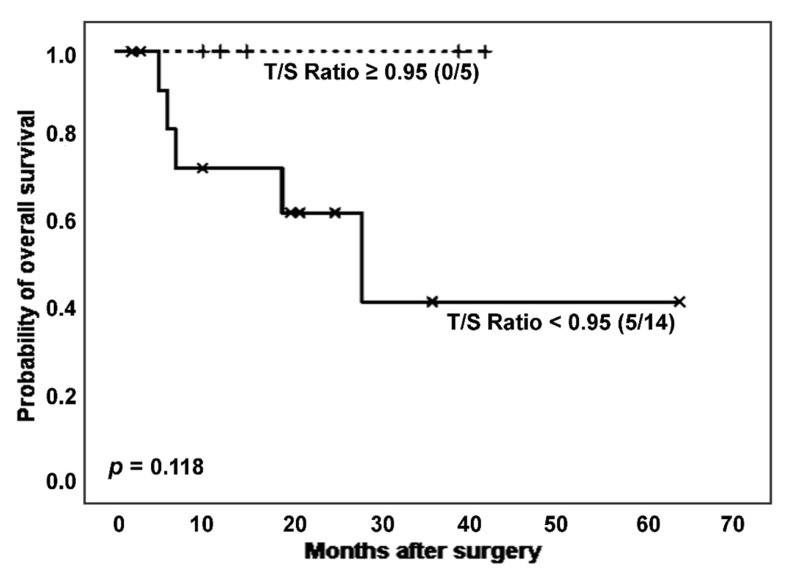
Survival curves of non-cancer IPF patients according to categorized T/S ratio.

**Figure 6 biomedicines-11-03257-f006:**
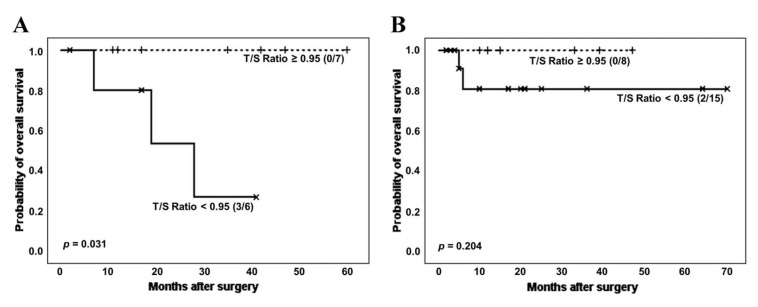
Survival curves of patients without cancer according to categorized T/S ratio and stratifying by the co-occurrence of ILD and RD: (**A**) patients with RD; (**B**) patients without RD.

**Table 1 biomedicines-11-03257-t001:** Clinico-pathological variables of patients included in this study.

Variable	Total ILD (n = 61)	IPF (n = 28)	Other ILDs (n = 33)	*p* Value ^1^
Mean follow-up, months (range)	23 (1–70)	22 (2–66)	24 (1–70)	0.587 ^2^
Gender				0.945 ^3^
Males, n (%)	46 (75.4)	21 (75.0)	25 (75.8)	
Females, n (%)	15 (24.6)	7 (25.0)	8 (24.2)	
Age, mean (SD)	64.97 (11.56)	69.0 (6.40)	61.5 (13.77)	0.008 ^4,^*
Cancer, n (%)	21 (34.4)	8 (28.6)	13 (39.4)	0.375 ^3^
Rheumatic disease (RD), n (%)	24 (39.3)	10 (35.7)	14 (42.4)	0.593 ^3^
Tobacco exposure, n (%)	43 (70.5)	24 (85.7) *	19 (57.6)	0.016 ^3,^*
Drug exposure, n (%)	11 (18)	5 (17.9)	6 (18.2)	0.974 ^3^
Dust exposure, n (%)	16 (26.2)	8 (28.6)	8 (24.2)	0.702 ^3^
Occupational exposure, n (%)	24 (39.3)	10 (35.7)	14 (42.4)	0.593 ^3^
COVID-19 exposure, n (%)	11 (18)	6 (21.4)	5 (15.2)	0.525 ^3^

ILD: interstitial lung disease; IPF: idiopathic pulmonary fibrosis; SD: standard deviation; ^1^ For the comparison between “IPF” and “other ILDs” categories; ^2^ Mann–Whitney U test; ^3^ Chi-squared test; ^4^ Student’s *t*-test; * Statistically significant for the comparison at *p* < 0.05.

**Table 2 biomedicines-11-03257-t002:** Univariate and multivariate hazard Cox regression for the risk of death.

Variable		Non-Adjusted HR (95% CI)	Unadjusted *p* Value	Adjusted HR (95% CI)	Adjusted *p* Value
Age		1.06 (1.00 to 1.12)	0.044	1.08 (0.996 to 1.17)	0.063
IPF	Absent	1.0 (reference)	-	1.0 (reference)	-
	Present	3.49 (1.09 to 11.2)	0.035	4.43 (1.23 to 15.9)	0.023
RD	Absent	1.0 (reference)	-	1.0 (reference)	-
	Present	2.67 (0.87 to 8.18)	0.085	608.8 (5.71 to 64885.0)	0.007
T/S ratio		0.58 (0.07 to 4.68)	0.607	3.53 (0.25 to 49.8)	0.350
T/S ratio x RD		0.016 (0.00 to 2.19)	0.099	0.002 (0.00001 to 0.54)	0.029

CI: confidence interval; HR: hazard ratio; IPF: idiopathic pulmonary fibrosis; RD: rheumatic disease; x:interaction

**Table 3 biomedicines-11-03257-t003:** Inflammatory parameters analyses in patients without cancer, considering IPF and RD co-occurrence.

Inflammatory Parameter	IPF and RD n = 6	IPF n = 14	*p* Value ^1^
T/S ratio, mean (SD)	0.739 (0.45)	0.744 (0.37)	0.853
Fibrinogen (mg/L), mean (SD)	413.5 (79.84)	458.1 (96.1)	0.599
CRP (mg/L), mean (SD)	3.30 (3.78)	9.51 (12.74)	0.335
ESR (mm/h), mean (SD)	59.5 (71.4)	23.6 (29.15)	0.439
Ferritin (mg/L), mean (SD)	5143.5 (6708.32)	169.2 (101.31)	0.032 *

CRP: C-reactive protein; ESR: erythrocyte sedimentation rate; IPF: idiopathic pulmonary fibrosis; RD: rheumatic disease; SD: standard deviation; ^1^ Mann–Whitney U test; * Statistically significant for the comparison at *p* < 0.05.

## Data Availability

The datasets used and/or analyzed during the current study are available from the corresponding author on reasonable request.

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
