# Peer review of "Role of Telomere Length in Survival of Patients with Idiopathic Pulmonary Fibrosis and Other Interstitial Lung Diseases"

_biomedicines, 2023, doi:10.3390/biomedicines11123257_

Round 1

Reviewer 1 Report

Comments and Suggestions for Authors

The authors review interstitial lung diseases (ILD), particularly idiopathic pulmonary fibrosis (IPF). They investigated the role of telomeres (TL) as a prognostic factor in patients with ILD and found that in patients with ILD, particularly in cases with co-occurrence of RD and IPF, TL could be considered a risk-modifying variable for death. The manuscript is well written, in standart English. The correlative analysis made perfectly, and at the same time, it makes the readability of the article difficult.

Minor comments:

1. row 58....."Several mediators are involved in the..." to be modified.

2. Figure 1 is boring, to be modified. 

3. Figure 6 "A and B" parts to be unified.

4. The row 287 "Nevertheless, considering that lung tissue of pa- 287 tients..." to be modified. 

5. The row 335 "We hypothesized that a higher inflammatory state could be the rationale for 336 the performance of TL as a modifying variable.." to be modified. 

6. The authors examined oxidative disorders directly related to the role of MUC5B, MMP7, MMP9, CXCL10 and IL-23. If - yes, then probably an additional paragraph with confirming comments and helped the research even better.

7. The conclusion part is need to be modified. 

8. The development of an introductory scheme after the abstract would support the topicality and relevance of the subject.

Comments on the Quality of English Language

Minor editing of English language required

Author Response

Responses to the Review Report (Reviewer 1):

- Modifications suggested by the reviewer in points 1, 4 and 5 have been included in the revised version of the manuscript (rows 58-59, 288-289, and 336-338).

- Figure 1 has been eliminated in the revised version of the manuscript, since its content has been explained in the material and methods section of the manuscript. Instead, we have included an introductory scheme (page 2 of the revised version of the paper), as reviewer 1 suggested.

- Figure 6 “A and B” parts have been unified, as reviewer 1 indicated (in the revised version, Figure 5).

- The conclusion part has been modified, as reviewer 1 suggested (rows 365-371).

- We confirm that we have not examined oxidative disorders directly related to the role of MUC5B, MMP7, MMP9, CXCL10 and IL-23.

- English editing language has been revised.

Reviewer 2 Report

Comments and Suggestions for Authors

Dear Authors!

Thank you for the opportunity to review your manuscript.

The IPF is a severe disease with multiple pathogenetic features. The idea to find the telomer length in chronic lung disease is new and promising. The introduction is describe the actuality of the problem. Authors provide the clear and reproducible results, explain the main idea of the study and discussion contains all contemporary literature.

I have only few suggestion

Table: Rheumatological. Better use term rheumatic.

Can you provide in the demography the spectrum of the rheumatic disease and\or type of autoantibodies in the patients

Can you provide the additional regression model adding more predictors, not only telomers length (this is optional suggestion)

Author Response

Responses to the Review Report (Reviewer 2):

- The term “Rheumatological” has been changed by “rheumatic” along the manuscript, such as reviewer 2 suggested.

- The spectrum of the rheumatic disease has been included in row 230 (revised version of the manuscript), attending reviewer 2 suggestion.

- Finally, with respect to the last suggestion of the reviewer 2, in order to run a Cox regression model, the choice of an adequate sample size is generally based on the rule of thumb derived from simulation studies of a minimum of 10 events per variable. Because of a limited number of events, adding more variables would diminish statistical significance of some predictors or the validity and, therefore, the interest of our Cox regression model. To overcome this limitation, we performed a variable selection process, which yielded the most significant predictors to be included in a final model. Anyway, we have run the analysis you suggest (data not published), adding cancer, tobacco exposure and age x T/S ratio interaction, and our results point in the same direction (the variables introduced are not significant) but the p-value of the remaining predictors are increased.